# Phytochrome-interacting factors directly suppress *MIR156* expression to enhance shade-avoidance syndrome in *Arabidopsis*

Yurong Xie[1], Yang Liu[1], Hai Wang[1], Xiaojing Ma[1,2], Baobao Wang[1], Guangxia Wu[1] & Haiyang Wang[1]

Plants have evolved a repertoire of strategies collectively termed the shade-avoidance syndrome to avoid shade from canopy and compete for light with their neighbors. However, the signaling mechanism governing the adaptive changes of adult plant architecture to shade is not well understood. Here, we show that in *Arabidopsis*, compared with the wild type, several PHYTOCHROME-INTERACTING FACTORS (PIFS) overexpressors all display constitutive shade-avoidance syndrome under normal high red to far-red light ratio conditions but are less sensitive to the simulated shade, whereas the *MIR156* overexpressors exhibit an opposite phenotype. The simulated shade induces rapid accumulation of PIF proteins, reduced expression of multiple *MIR156* genes, and concomitant elevated expression of the *SQUA-MOSA-PROMOTER BINDING PROTEIN-LIKE (SPL)* family genes. Moreover, in vivo and in vitro assays indicate that PIFs bind to the promoters of several *MIR156* genes directly and repress their expression. Our results establish a direct functional link between the phytochrome-PIFs and miR156-SPL regulatory modules in mediating shade-avoidance syndrome.

[1] Biotechnology Research Institute, Chinese Academy of Agricultural Sciences, Beijing 100081, China. [2] Graduate School of Chinese Academy of Agricultural Sciences, Beijing 100081, China. Yurong Xie and Yang Liu contributed equally to this work. Correspondence and requests for materials should be addressed to H.W. (email: wanghaiyang@caas.cn)

By the year 2050, the world population is expected to reach 9.3 billion and it is a daunting task to produce enough food, feed, and biofuel materials with limited resources[1]. Over the past few decades, increasing planting density has been an effective way of improving crop yields per unit land area[2]. However, a key factor that limits planting density in modern agricultural practice is the plant's shade-avoidance response, which is triggered when plants detect a reduction of red (R) to far-red (FR) light ratios (R: FR) in their environment due to absorption of R light by neighboring plants. Typical shade-avoidance responses include increased plant height, elevated leaf angles to horizontal, reduced branching, decreased leaf blade area, and early flowering[3]. These traits are collectively called shade-avoidance syndrome (SAS)[4]. Although it is deemed that a robust SAS allows the plants to compete with the neighbors for limited resources, and to complete their life cycle earlier to ensure a reproductive success such plasticity comes at a cost of reduced fitness[5]. For crop species, the SAS could cause a reduction in yields due to reduced investment of resources on reproductive development and decreased immunity to plant pests and pathogens[6, 7]. Thus, it is generally believed that SAS has largely been attenuated or refined during crop domestication and genetic improvement[8]. However, currently little is known regarding the molecular genetic mechanisms governing SAS in crops, which severely hampers our ability to breed high-density tolerant crop cultivars.

In the model dicot plant species Arabidopsis thaliana, SAS is primarily mediated by the photoreceptor phytochrome B (phyB), with phyD and phyE playing a minor role[9]. Arabidopsis phyB mutants exhibit a constitutive shade-avoidance response even under normal high R:FR conditions, including elongation of hypocotyl, petioles and stem, accelerated flowering, and increased apical dominance, indicating that phyB negatively regulates

SAS[10]. Phytochrome is synthesized within the cytosol in its inactive R light absorbing Pr form, and upon exposure to R light it can rapidly convert into its biologically active FR light-absorbing Pfr form. Reversion of Pfr to Pr form occurs in FR light-enriched environments or more slowly in the dark[11]. Thus, under shade conditions (low R:FR), the majority of phyB is inactivated, thus triggering a shade-avoidance response.

Previous studies have also identified numerous regulators of SAS acting downstream of the phytochrome photoreceptors in Arabidopsis. Among them, a family of PHYTOCHROME-INTERACTING FACTORS (PIFs, including PIF1, PIF3, PIF4, PIF5, and PIF7) is believed to play an instrumental role in mediating shade-induced rapid transcriptome reprogramming and subsequent SAS responses[12–15]. Among the PIFs-regulated genes, a dozen of PHYTOCHROME RAPIDLY REGULATED (PAR) genes (e.g., ATHB2, ATHB4, HFR1, PAR1, PAR2, PIL1, HAT1, HAT2, HAT3, BEE1, BIM1, BBX21, BBX22, BBX24, and BBX25) are believed to be essential for implementing the SAS responses. These factors either act positively or negatively to bring about a balanced SAS responses[16]. In addition, it has been shown that several PIFs (PIF4, PIF5, and PIF7) can directly regulate the expression of several auxin biosynthetic (TAA1 and YUC genes) or response genes (such as IAA29) to promote hypocotyl elongation[13, 15, 17]. However, it should be noted that most studies on SAS responses have been focused on the hypocotyl elongation process in Arabidopsis. A recent study combining RNA-seq with phenotypic profiling revealed that each shade-avoidance response (hypocotyl, petiole, and flowering time) was governed by shared and separate pathways, suggesting that cautions should be excised when generalizing conclusions from studies on hypocotyls[18]. Thus, our understanding of SAS molecular mechanisms remains incomplete and fragmented.

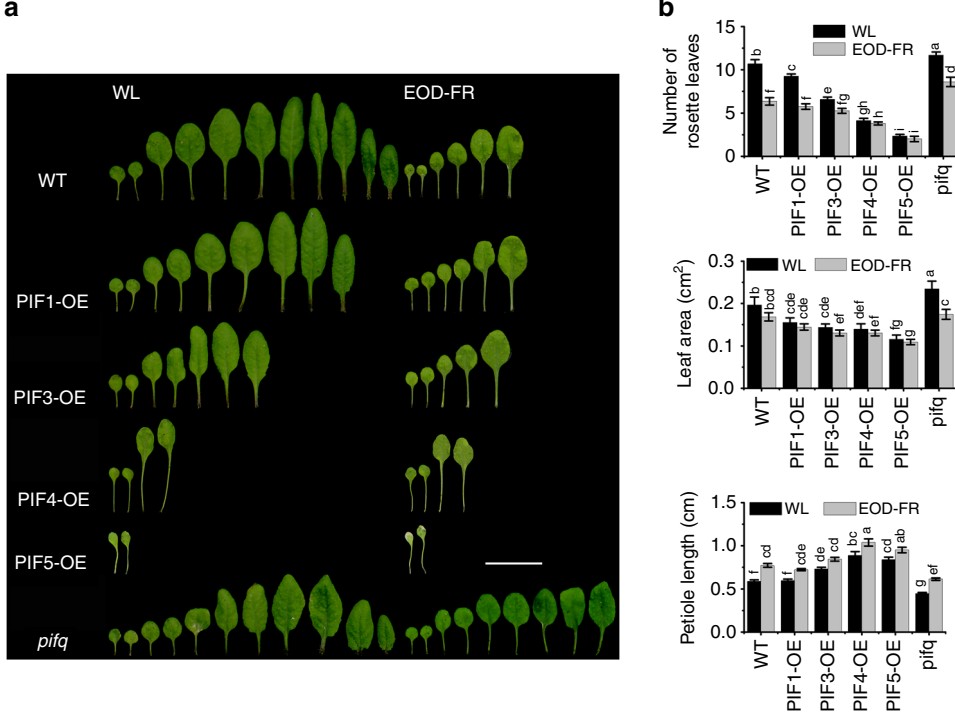

**Fig. 1** Adult *PIF* overexpressors exhibit constitutive SAS under normal high R:FR conditions and attenuated SAS responses under simulated shade conditions. **a** Comparison of the rosette leaf number, leaf blade area, and petiole length of the first and second leaf between *PIF* overexpressors, *pifq*, and WT plants grown under normal high R:FR (WL) or simulated shade (EOD-FR) conditions. Eight-day-old seedlings were moved into the soil and grown under WL with or without EOD-FR treatment for 4 weeks. Bar = 2 cm. **b** Quantification of the rosette leaf number, leaf blade area, and petiole length of *PIF* ovexpressors, *pifq*, and WT under WL or EOD-FR conditions. Values shown are mean ± SD (n = 12). Different letters indicate significant differences by two-way ANOVA

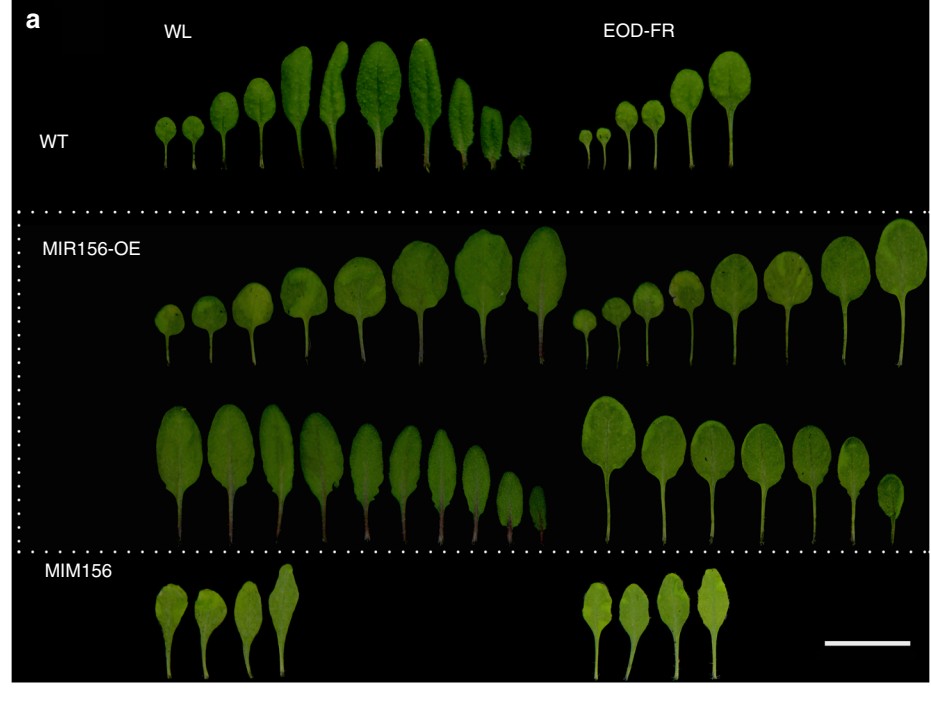

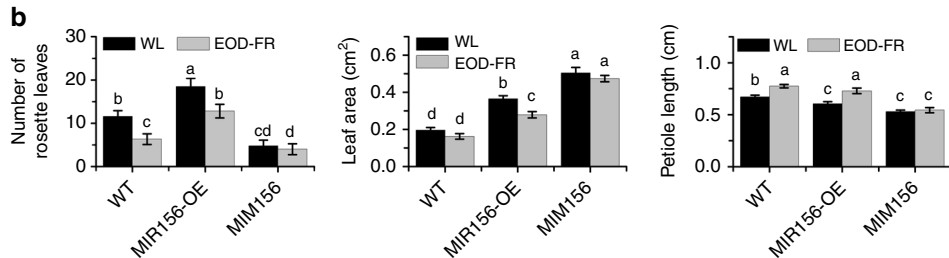

**Fig. 2** *MIR156* overexpression represses attenuated SAS responses in adult *Arabidopsis* plants. **a** Comparison of the rosette leaf number, leaf blade area, and petiole length of the first and second leaf between *MIR156*overexpressors, *MIM156*, and WT plants under normal high R:FR (WL) or simulated shade (EOD-FR) conditions. Eight-day-old seedlings were moved into the soil and grown under WL with or without EOD-FR treatment for 4 weeks. Bar = 2 cm. **b** Quantification of the rosette leaf number, leaf blade area, and petiole length of *MIR156* overexpressors, *MIM156*, and WT plants under WL or EOD-FR conditions. Values shown are mean ± SD ($n = 12$). Letters indicate significant differences by two-way ANOVA

Recent studies have shown that *MIR156s*, together with their downstream targets *SQUAMOSA-PROMOTER BINDING PROTEIN-LIKE* (*SPL*) family of genes, are pivotal regulators of various biological processes in plants, such as the timing of vegetative to reproductive phase transition, leaf development, branching/tillering, fruit ripening, fertility, and response to stresses[19, 20]. As many of the *MIR156s*-mediated developmental processes overlap with the phytochrome-mediated SAS responses, we investigated the regulatory relationship between the phy-PIFs signaling pathway and the miR156-SPL regulatory module in mediating SAS. We show that several PIF overexpressors (*35S::PIF1-OE*, *35S::PIF3-OE*, *35S::PIF4-OE*, and *35S::PIF5-OE*) all display enhanced SAS responses under normal high R:FR ratio conditions, and attenuated sensitivity to the simulated shade (including leaf number, leaf blade size, branches, plant height, petiole length, and flowering time), compared to the wild-type plants (WT). In addition, the accumulation of PIF proteins rapidly increases in response to simulated shade. Moreover, multiple PIFs (PIF1, PIF3, PIF4, and PIF5) can directly bind to the G-box motifs present in the promoters of several *MIR156* genes and down-regulate their expression. Our results provide a direct functional link between the phytochrome-PIFs and miR156-SPL regulatory modules in mediating SAS during plant vegetative growth and development.

## Results

**Adult *PIF* overexpressors exhibit constitutive SAS.** Previous studies showed that at the seedling stage, the shade-induced hypocotyl elongation is significantly compromised in the *pif4 pif5* double mutant, and to an even greater extent in the *pif1 pif3 pif4 pif5* quadruple (*pifq*) and *pif7* mutants[12, 14, 15]. Conversely, seedlings overexpressing *PIF4* and *PIF5* have constitutively long hypocotyls and petioles even under normal high R:FR conditions[12]. To examine the effects of *PIF* overexpression at the vegetative and adult stages, we obtained the previously generated *35S::PIF1-OE*, *35S::PIF3-OE*, *35S::PIF4-OE*, and *35S::PIF5-OE* transgenic plants[21–24], together with the WT control and *pifq* mutant, and grew them under normal high R:FR conditions. The blade area of the first and second leaf, length of the first and second petiole, number of rosette-leaf branches, number of rosette leaves at bolting, and plant height were measured. Our data showed that under high R:FR conditions, the numbers of both rosette leaves and rosette-leaf branches were reduced significantly in the *PIF* overexpressors, especially in the *35S::PIF4-OE* and *35S::PIF5-OE* lines. On average, the *35S::PIF-OE* lines had only ~21.8–86.5% rosette leaves and ~8.6–76.4% rosette-leaf branches, respectively, compared with the WT (Fig. 1a, b and Supplementary Fig. 1a, b). Also, the first and second petioles of the *35S::PIF-OE* lines were much longer (~24.1–52.3% longer

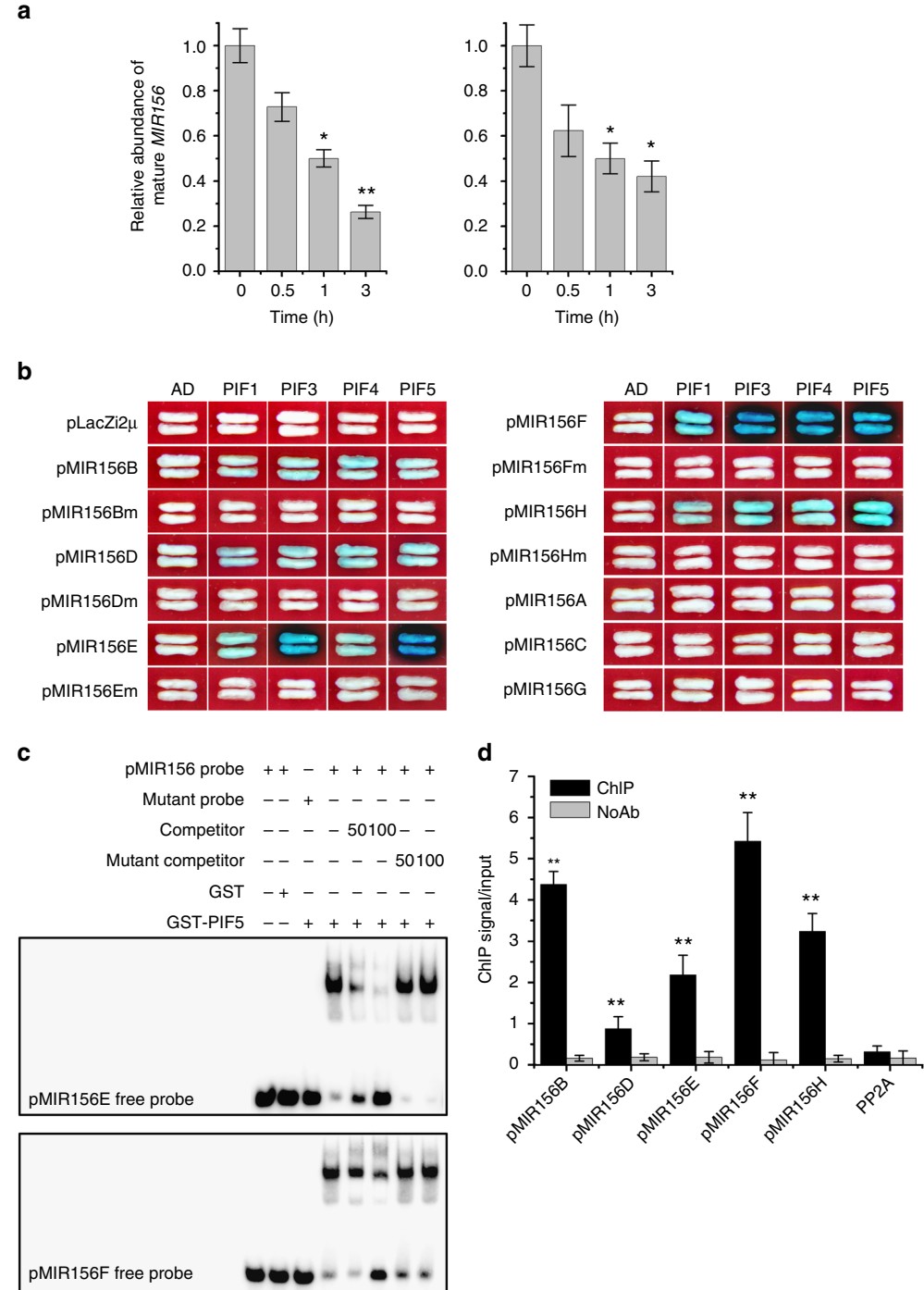

**Fig. 3** PIFs directly bind to *MIR156* promoters. **a** The transcript levels of mature *MIR156* decline in WT in response to EOD-FR treatment. Eight-day-old (*left panel*) or 3-week-old (*right panel*) WT plants grown under WL were treated with or without EOD-FR for 6 days before harvested for RNA extraction. Values given are mean ± SD (*n* = 3). *P < 0.05 and **P < 0.01 by the Student's *t*-test. **b** Y1H assay showing direct binding of PIFs to the PBE-box or G-box motifs in several *MIR156* promoters. The length and PIF-binding elements of each promoter fragment is shown in Supplementary Fig. 6. **c** EMSA showing that the GST-PIF5 bHLH domain recombinant protein binds to biotin-labeled probes of *MIR156E* (*up panel*) and *MIR156F* (*down panel*). **d** ChIP assay of *MIR156* in 2-week-old *35S::PIF5-HA* seedlings grown under high R:FR conditions and treated with EOD-FR for one time before harvesting. Values given are mean ± SD (*n* = 3). **P < 0.01 by the Student's *t*-test

than that of WT). In addition, the blade sizes of the first and second leaf were significantly smaller in the *35S::PIF-OE* lines compared with the WT (~59.1–78.8% of that of WT) (Fig. 1a, b). At maturity, all *35S::PIF-OE* lines (except *PIF1-OE* line) were taller than the WT and all *35S::PIF-OE* lines flowered earlier than WT when measured by the number of rosette leaf at bolting

(Supplementary Fig. 1a, b). In contrast, the *pifq* mutant displayed a largely opposite phenotype to the *35S::PIF-OE* lines (had slightly more rosette leaves, larger first and second leaves, shorter first and second petioles) (Fig. 1a, b). These observations indicate that these PIFs play a positive role in regulating various aspects of SAS at the vegetative and adult stages.

**Adult PIF overexpressors are less sensitive to shade**. Since FR light treatment at the end-of-day (EOD-FR) causes similar plant phenotypic changes to those grown under canopy shade (reduced R:FR ratios)[4], thus EOD-FR treatment was adopted in this study to investigate the response of PIF-related materials to mimic shade conditions. The 35S::PIF-OE lines, pifq mutant, and WT plants were first germinated and grown under normal high R:FR (white light (WL)) conditions for 1 week, and then treated with EOD-FR for 15 min each day before returning to darkness. After EOD-FR treatment for 4 weeks, the number of rosette leaves and rosette-leaf branches, blade sizes of the first and second leaf, and lengths of the first and second petiole were measured. Compared to their counterparts grown under normal R:FR conditions, EOD-FR-treated WT, pifq, 35S::PIF1-OE, and 35S::PIF3-OE plants all had significantly reduced rosette leaf number, while the rosette leaf number of 35S::PIF4-OE and 35S::PIF5-OE declined slightly (Fig. 1b). Also the number of rosette-leaf branches and the blade areas of the first and second leaf were reduced, but the lengths of the first and second petioles were increased in the EOD-FR-treated plants (Fig. 1b and Supplementary Fig. 1b). Notably, under EOD-FR treatment, the 35S::PIF-OE plants had milder reductions (percentages) in leaf number, rosette leaf branches, the first and second leaf size, and only moderate increases in petiole length and plant height compared with the WT plants. In contrast, pifq plants had larger reductions (percentages) in rosette leaf branches and the first and second leaf size, and larger increases in petiole length and plant height under EOD-FR treatment compared with the WT (Fig. 1b and Supplementary Fig. 1b). These observations suggest that the 35S::PIF-OE lines were less sensitive, whereas the pifq mutant was more sensitive to the EOD-FR treatment compared with the WT.

**MIR156 overexpressors exhibit attenuated SAS responses**. Arabidopsis MIR156B overexpressing plants were reported to have more rosette leaves, increased branching (including axillary, quaternary, and quinary branches), prolonged expression of juvenile vegetative traits, and late flowering when grown under normal high R:FR conditions[25, 26]. To investigate a possible role of MIR156s in shade-avoidance response, we generated over-expressors for several MIR156 genes (including MIR156B, MIR156D, MIR156E, MIR156F, and MIR156H) and they all showed similar phenotypes including more rosette leaves and late flowering (Supplementary Fig. 2a, b). In this study, MIR156B overexpressor (designated MIR156-OE hereafter) was used for further detailed investigation. When grown under our normal high R:FR conditions, MIR156-OE produced ~59.6% more rosette leaves, ~35.2% more rosette-leaf branches, and ~86.4% bigger leaves, but obviously shorter height (only ~51.3% of WT) and slightly shorter petioles (~89.9% of WT) compared with the WT (Fig. 2a, b and Supplementary Fig. 3a, b). On the contrary, the MIM156 transgenic line, which has significantly reduced MIR156 expression[27], displayed a largely opposite phenotype: fewer rosette leaves (~41.2% of WT) and rosette-leaf branches (~68.5% of WT), significantly shorter petioles (~78.6% of WT) and much bigger first and second leaves (~157.4% of WT) (Fig. 2a, b and Supplementary Fig. 3a, b).

To test the role of MIR156s on SAS, both MIR156-OE and MIM156 lines were treated with EOD-FR. The results showed that EOD-FR treatment significantly reduced the rosette leaf number in both WT and MIR156-OE line (~45.4 and ~30.76% reduction for WT and MIR156-OE, respectively). EOD-FR treatment also lead to reduced rosette-leaf branches (~54.9 and ~57.8% reduction for WT and MIR156-OE, respectively), smaller blade area for the first and second leaf (~17.3 and ~23.7% reduction for WT and MIR156-OE, respectively) compared with

their respective counterparts grown under high R:FR (Fig. 2b and Supplementary Fig. 3b). The MIR156-OE line also had greater increase in plant height (~26.1 and ~41.1% for WT and MIR156-OE, respectively) and the first and second petiole length (~15.2 and ~21.6% for WT and MIR156-OE, respectively) in response to the EOD-FR treatment compared to the WT (Fig. 2b and Supplementary Fig. 3b). Together, these results suggest that the MIR156 overexpressors exhibit reduced SAS under normal high R:FR ratio conditions, and are more sensitive to the simulated shade.

**The transcript levels of MIR156s decrease under shade**. Previous studies have shown that under high R:FR conditions, PIF1, PIF3, PIF4, PIF5, and PIF7 proteins all physically interact with phyB through their conserved N-terminal sequence, which in turn led to rapid phosphorylation and subsequent degradation of these PIF proteins by the 26S ubiquitin-proteasome pathway. This provides an elegant mechanism to rapidly regulate nuclear gene expression in response to the changing light environment[28]. In accordance with this model, PIF3, PIF4, and PIF5 proteins accumulate rapidly upon exposure to low R:FR[12, 14]. To test the effect of simulated shade on PIF protein accumulation, we conducted a time course immunoblot analysis of PIF with 2-weeks-old 35S::PIF1-OE, 35S::PIF3-OE, 35S::PIF4-OE, and 35S::PIF5-OE seedlings exposed to EOD-FR treatment. We observed an increased accumulation of these PIF proteins in seedlings treated with EOD-FR for 30 min, 1 h, and 3 h after EOD-FR treatment, compared to seedlings grown under normal WL conditions (Supplementary Fig. 4a). Quantitative reverse transcriptase (RT)-PCR assay showed that expression of several shade-avoidance marker genes (PIL1, HFR1, ATHB2, CKX5, XTR7, and IAA19)[14, 29] was obviously increased in PIF overexpressors (Supplementary Fig. 4b) and significantly upregulated by the EOD-FR treatment (Supplementary Fig. 4c), thus validating the effectiveness of our EOD-FR treatment.

We next examined the mature MIR156 RNA levels in plants treated with simulated shade. Quantitative RT-PCR analysis showed that the total mature MIR156 RNA level was rapidly and significantly downregulated in 2-weeks-old Arabidopsis seedlings or 4-weeks-old adult plants treated with EOD-FR (Fig. 3a). To confirm this, we examined the primary transcripts of MIR156B, MIR156D, MIR156E, MIR156F, and MIR156H in response to EOD-FR treatment. As expected, the primary transcript levels of these MIR156s all declined significantly when exposed to EOD-FR (Supplementary Fig. 5a). To further verify this, we generated transgenic plants harboring the β-glucuronidase (GUS) reporter gene driven by the MIR156B, MIR156D, MIR156E, MIR156F, and MIR156H promoter, respectively. The pMIR156::GUS reporter gene activities were all similarly downregulated by exposure to EOD-FR (Supplementary Fig. 5b, c). In addition, we detected the expression of several miR156-targeted SPL genes and found most of them were obviously upregulated when exposed to EOD-FR (Supplementary Fig. 5d).

**PIFs directly bind to G-box motifs in MIR156 promoters**. PIFs and MIR156s are positive and negative regulators of SAS, respectively, and the abundance of MIR156 decreases upon PIF protein accumulation, suggesting that PIFs may directly suppress MIR156s expression. To test this hypothesis, we first analyzed the cis-elements of the putative promoters (~3 kb upstream of the mature MIR156 sequence) of all eight Arabidopsis MIR156 members (MIR156A-H) for PIF-binding sites. One or multiple typical PIF-binding sites (G-box or PBE-box) were found in all MIR156 promoters (Supplementary Fig. 6). Yeast one-hybrid assay showed that all four PIFs tested here (PIF1, PIF3, PIF4, and

PIF5) could bind to the G-box motifs present in the promoters of *MIR156B*, *MIR156D*, *MIR156E*, *MIR156F*, and *MIR156H*, with PIF3 and PIF5 showing the strongest binding to the *MIR156E* and *MIR156F* promoters. Mutations of the G-box sequences in their promoters abolished the binding, suggesting that the binding was specific. No obvious binding of PIFs to the promoters of

*MIR156A*, *MIR156C*, or *MIR156G* was detected in this assay (Fig. 3b and Supplementary Fig. 7). To confirm this binding, we generated recombinant protein of the bHLH DNA-binding domain of PIF5 (Supplementary Fig. 8) and used it in a gel electrophoresis mobility shift assay (EMSA) with the *MIR156E* and *MIR156F* promoter fragments containing the G-box. As

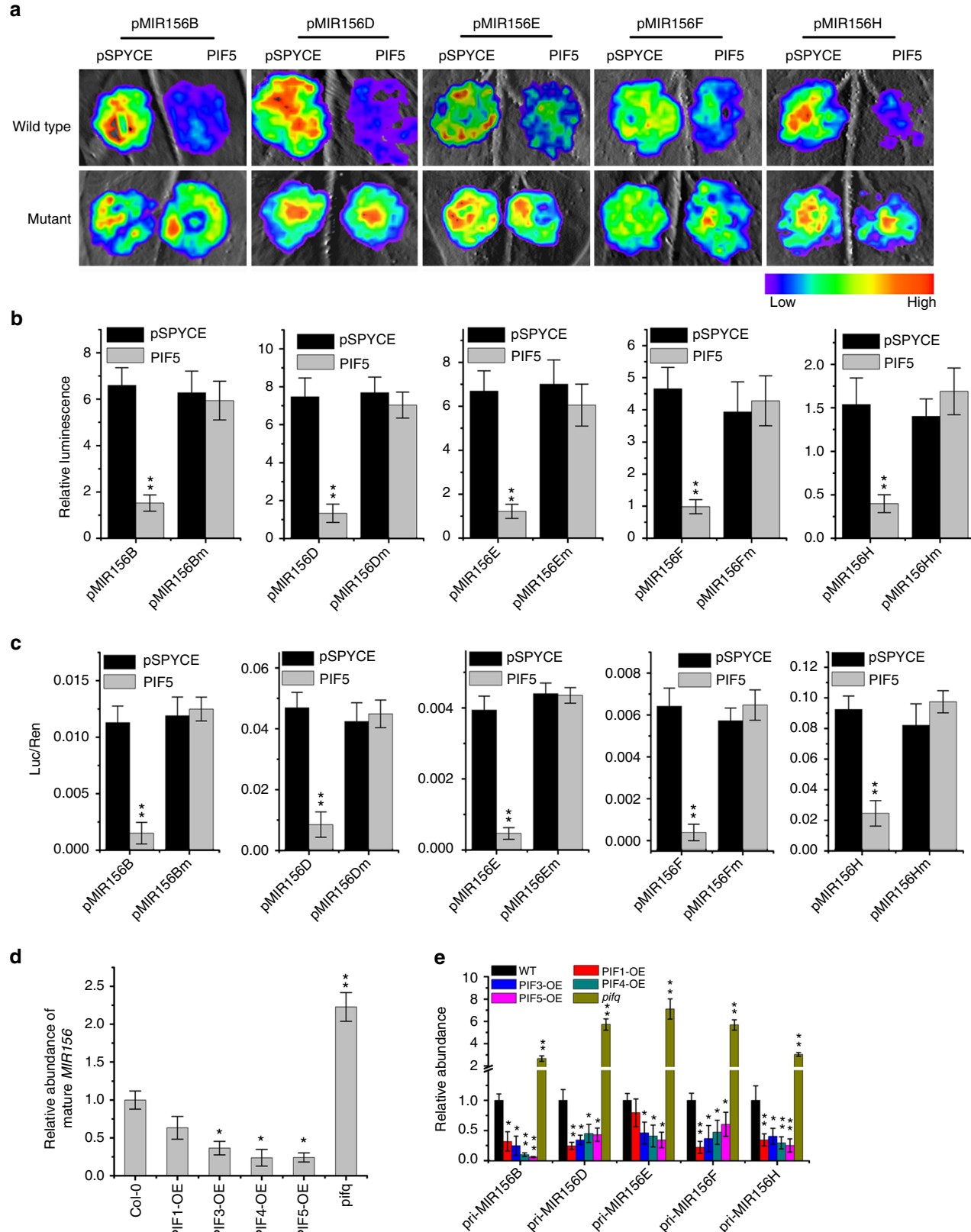

shown in Fig. 3c, PIF5 specifically bound to the promoter fragments containing normal G-box sequence, but not the promoter fragments with mutated G-box sequence. Further, our chromatin immunoprecipitation (ChIP)-PCR assay using the PIF5-HA transgenic plants confirmed enrichment of *MIR156B*, *MIR156D*, *MIR156E*, *MIR156F*, and *MIR156H* promoter fragments containing the G-box motif (Fig. 3d). These results verified the in vivo binding of PIF5 to multiple members of the *MIR156* gene family and confirmed an earlier report that *MIR156E* (AT5G11977) was identified as a putative direct target gene of PIF5[13].

**PIFs directly repress the expression of several *MIR156* genes.** We next performed a transient expression assay to examine whether PIF proteins could directly regulate the transcription of the target *MIR156s*. Agrobacterium strains harboring the *p35S:PIF* effector and *pMIR156:LUC* reporter plasmids were injected into *Nicotiana benthamiana* leaf epidermal cells. After incubation in darkness at 25 °C for 2–3 days, the *pMIR156:LUC* reporter gene activity was examined. Co-expression of PIF protein strongly inhibited the *LUC* reporter gene activities driven by the endogenous promoters of *MIR156B*, *MIR156D*, *MIR156E*, *MIR156F*, and *MIR156H*, but this inhibition was abolished by mutations in the G-box motifs in these promoters (Fig. 4a–c and Supplementary Fig. 9). Consistent with this, most primary *MIR156* transcripts and the total mature *MIR156* levels were lower in the *35S::PIF-OE* lines, but significantly increased in the *pifq* plants compared to the WT (Fig. 4d, e). Further, the expression of *SPL* genes, downstream target genes of miR156, increased in the *35S::PIF-OE* lines (Supplementary Fig. 10). We thus concluded that PIF5 (and most likely other PIFs as well) could bind to the promoters of multiple *MIR156* members directly and repress their expression.

**_MIR156s_ act downstream of _PIFs_ in regulating SAS.** To further investigate the possible genetic relationship between *PIF* and *MIR156s* in regulating SAS, we generated *PIF5-OE/MIR156-OE*, *pifq/MIR156-OE*, *PIF5-OE/MIM156*, and *pifq/MIM156* plants via genetic crosses and examined their responses to simulated shade treatment. As expected, plants of the *pifq/MIR156-OE* genotype displayed a similar phenotype to that of *MIR156-OE* line in the number of rosette leaves, plant height, number of rosette-leaf branches, length of the first and second petioles, and the size of the first and second leaves (Fig. 5 and Supplementary Fig. 11). However, *PIF5-OE/MIR156-OE* displayed fewer rosette leaves and rosette-leaf branches, longer petioles, and elevated plant heights than *MIR156-OE*, but much more rosette leaves and rosette-leaf branches and shorter petiole length and plant heights than *35S::PIF5-OE* (Fig. 5 and Supplementary Fig. 11). No significant phenotypic differences were detected among the *MIM156*, *PIF5-OE/MIM156*, and *pifq/MIM156* plants (Fig. 5 and Supplementary Fig. 11). These observations, together with the molecular data presented above, support the placement of *MIR156* downstream of PIF5 (and most likely other PIFs as well) in regulating various aspects of SAS.

**Discussion**

SAS refers to a collection of important adaptive morphological and physiological changes in plants that occur in response to reduced R:FR ratios caused by neighboring vegetation. Although this response is deemed to improve the reproductive success and thus the survival rates of the plants, it is detrimental to agricultural production of most cereal crops. Thus, it is generally believed that SAS has largely been attenuated or refined during crop domestication and genetic improvement[8, 30]. Indeed, there is evidence that phytochromes might have been subjected to selection during crop domestication and breeding, and attempts have been made to suppress SAS for better agronomic performance in several crops via manipulating the light signaling pathways through a transgenic approach. However, despite promising, these efforts have made limited success due to the associated pleiotropic negative effects of the transgenes[1, 31, 32]. As pressure on arable land persists, breeding of high planting density tolerant cultivars will continue to be a major objective in the years ahead. A better understanding of the signaling mechanisms governing SAS will definitely help meet this need.

In this study, we showed that in *Arabidopsis thaliana*, PIFs, a group of signaling integrators between light and other signals (various hormones, temperature, sugar, and so on)[33] directly bind to and repress the expression of several *MIR156* genes (*MIR156B*, *MIR156D*, *MIR156E*, *MIR156F*, and *MIR156H*) in response to simulated shade, causing altered developmental programs in multiple plant organs, and thus plant architecture (including leaf production and expansion, petiole and stem elongation, branches, and flowering time). It is well known that miR156 represents one of the most evolutionarily conserved miRNAs in plants[34]. Among the 17 *Arabidopsis SPLs*, 11 are miR156 targets, and these *SPLs* have been shown to regulate diverse developmental processes[20, 35]. Based on our results, we propose a putative model in which the FR-inactivated phytochromes (mainly phyB) induce a rapid stabilization and increased accumulation of several PIF proteins (PIF1, PIF3, PIF4, and PIF5), which in turn directly bind to the G-box motifs in the promoters of multiple *MIR156* genes and repress their expression, thus alleviating the inhibitory effect of miR156 on the downstream *SPL* genes. As transcription factors, the activated *SPL* genes then regulate diverse morphological changes associated with shade-avoidance responses via altering diverse sets of further downstream genes (Fig. 5c). Further studies are required to substantiate this proposition.

Notably, a recent study reported that in *Arabidopsis*, the transcriptional repressors DELLA proteins physically interact with SPL9 and interfere with its transcriptional activity, causing delayed floral transition by repressing miR172 in leaves and MADS-box genes at shoot apex, thus establishing a link between gibberellin-mediated and miR156/SPL module-mediated flowering pathways[36]. It has also been reported that DELLA proteins can inhibit PIFs proteins (PIF3 and PIF4) by sequestering their DNA-binding domains, thus antagonistically regulating hypocotyl elongation in plants[24, 37]. Further, a recent study showed that DELLA proteins can also accelerate degradation of PIF proteins by the 26S ubiquitin-proteasome pathway, suggesting a dual regulatory mechanism of PIFs by DELLAs[38]. However, no direct

**Fig. 4** PIF5 directly represses the expression of *MIR156* genes. **a** Transient expression assays shows that PIF5 directly represses the expression of *MIR156B*, *MIR156D*, *MIR156E*, *MIR156F*, and *MIR156H*. Representative images of *N. benthamiana* leaves 72 h after infiltration were shown. **b** Quantitative analysis of luminescence intensity in **a**. Five independent determinations were assessed. Values shown are mean ± SD ($n = 5$). **$P < 0.01$ by the Student's $t$-test. **c** Dual-luciferase assay of pMIR156::LUC expression. The expression of REN was used as an internal control. LUC/REN ration represents the relative activity of the *MIR156* promoter. Values given are mean ± SD ($n = 3$). **$P < 0.01$ by the Student's $t$-test. **d** Transcript levels of mature *MIR156* analyzed by qRT-PCR. Eight-day-old WT, *PIF* overexpressors, and *pifq* seedlings grown under WL were harvested for RNA extraction. Values given are mean ± SD ($n = 3$). *$P < 0.05$ and **$P < 0.01$ by the Student's $t$-test. **e** The relative expression levels of individual primary *MIR156* in 8-day-old WT, *PIF* overexpression, and *pifq* seedlings grown under high R:FR conditions. Values given are mean ± SD ($n = 3$). *$P < 0.05$ and **$P < 0.01$ by the Student's $t$-test

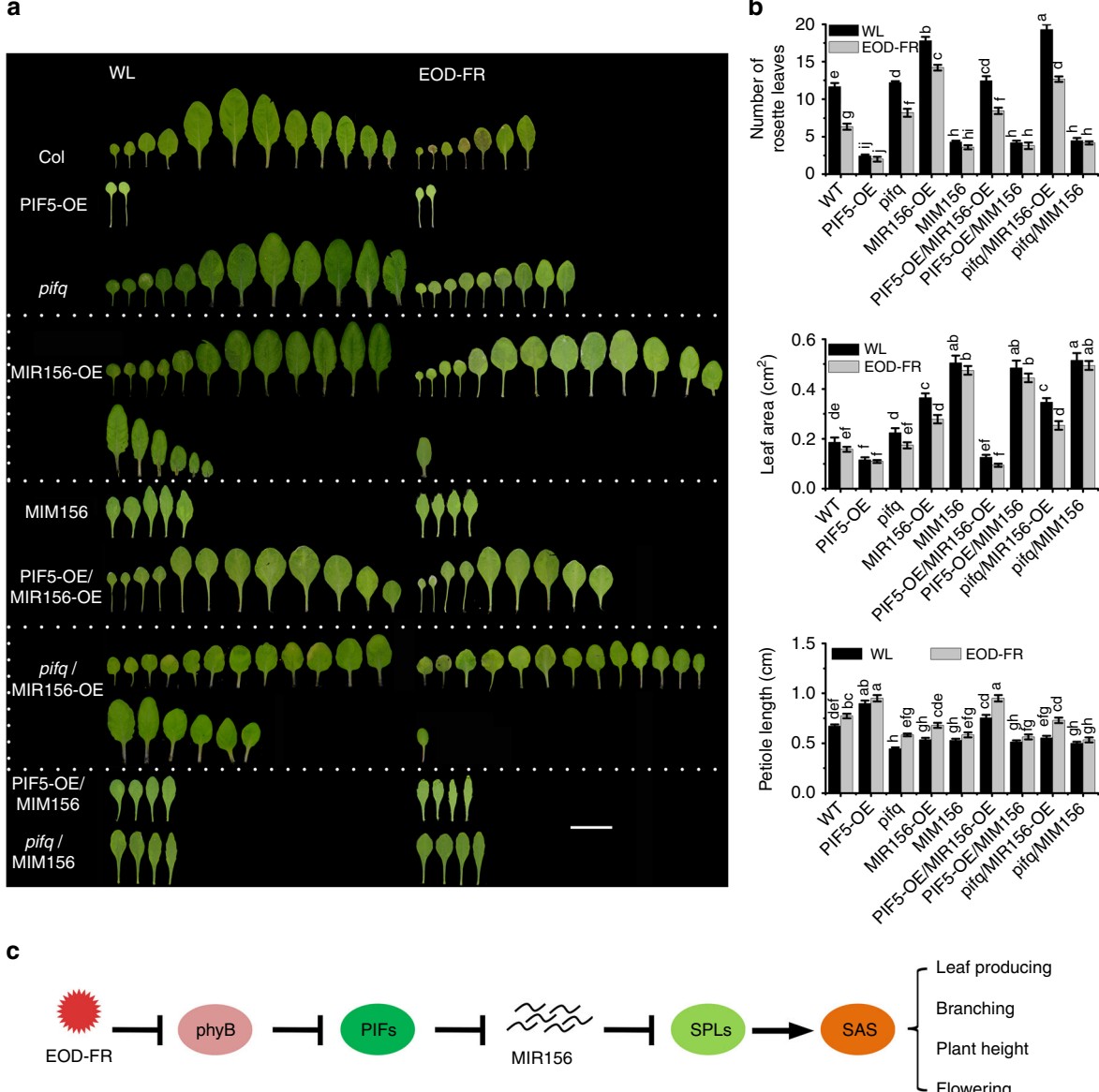

**Fig. 5** *MIR156s* act downstream of *PIFs* in regulating SAS. **a** The rosette leaf number, leaf blade area, and petiole length of the first and second leaf of WT, *PIF5-OE*, *pifq*, *MIR156-OE*, *MIM156*, and their higher order mutants grown under WL with or without EOD-FR treatment. Eight-day-old seedling grown on Murashige and Skoog agar medium under WL were transferred into soil and treated with EOD-FR for 4 weeks before phenotypic analysis. Bar = 2 cm. **b** Quantification of the rosette leaf number, leaf blade area, and petiole length of WT, *PIF5-OE*, *pifq*, *MIR156-OE*, *MIM156*, and their higher order mutants grown under WL with or without EOD-FR treatment. Values given are mean ± SD (*n* = 12). Letters indicate significant differences by two-way ANOVA. **c** Simplified schematic model depicting the signaling pathway of *PIFs* and *MIR156* to modulate shade-avoidance response in adult *Arabidopsis* plants. Shade (low R:FR ratios) inactivates phyB and induces a rapid accumulation of PIF proteins. These PIFs then directly bind to the promoters of multiple *MIR156* genes and repress their expression, which alleviates the inhibitory effect of MIR156s on their target *SPL* genes. The activated *SPL* genes then regulate diverse morphological changes associated with shade-avoidance responses via altering distinct sets of further downstream genes. *Arrow*: activate; *Bar*: repress

physical interaction between PIFs (PIF1, PIF3, PIF4, and PIF5) and SPLs (including SPL2, SPL3, SPL4, SPL5, SPL9, SPL10, SPL11, and SPL15) was detected in our yeast two-hybrid assay (Supplementary Fig. 12). Also no direct binding of PIFs to the promoters of above *SPLs* was detected in our yeast one-hybrid assay (Supplementary Table 2 and Supplementary Fig. 13). These results together suggest that PIFs regulate SPL activity mainly through regulating *MIR156* expression.

Accumulating evidence shows that the miR156/SPL regulatory module is highly conserved in land plant species, and plays important roles in regulating diverse plant developmental

processes and crop architecture[20]. Examples include the maize *TGA1* controlling glume development[39], maize *LG1* controlling leaf and tassel branch angle[40, 41], maize *UB2*, *UB3*, and *TSH4* regulating tassel and ear development, as well as vegetative tillering[42–44], rice *OsSPL16/GW8* and *OsSPL13/GLW7* controlling grain size[45, 46], rice *OsSPL14/IPA1/WFP* controlling plant architecture[47, 48], rice *OsSPL7* and *OsSPL17* regulating tillers and panicle architecture[49], and rice *OsLG1* controlling leaf and panicle branch angle[50, 51]. Notably, many of these genes are likely targets of selection during crop domestication or breeding for improved plant architecture and agronomic performance. In such

a context, it will be highly worthy to investigate whether the PHY-PIFs-miR156-SPLs regulatory circuitry identified in *Arabidopsis* also operates in crops, and how the components of this circuitry can be tailored to achieve desirable plant architecture and enhance yield per unit land area.

## Methods

**Plant materials and growth conditions.** All *Arabidopsis* materials used in this study are of the Columbia (Col-0) genetic background. The *PIF1-OE* (*35S::PIF1-MYC*) and *PIF3-OE* (*35S::PIF3-MYC*) transgenic lines were kindly provided by Dr. Rongcheng Lin[21, 22]. The *PIF4-OE* (*35S::PIF4-HA*) and *PIF5-OE* (*35S::PIF5-HA*) transgenic lines were supplied by Dr. Jiaqiang Sun[23, 24]. The stock seeds of *MIR156-OE* (*35S::MIR156*)[25] and *MIM156* (*35S::MIM156*)[52] were purchased from the European *Arabidopsis* Stock Centre (http://Arabidopsis.info).

*Arabidopsis* seeds were sown on Murashige and Skoog solid medium containing 1% sucrose after surface sterilization and vernalized at 4 °C in the dark for 3 days. Seedlings were then placed in a growth chamber (Percival, USA) and grown under continuous WL conditions at 23 °C (16 h light/8 h dark) for 8 days before further treatment or transferring into soil.

Tobacco (*Nicotiana benthamiana*) seeds were directly sown into the soil and grown in the culture room under WL conditions (16 h light/8 h dark) at 25 °C for 1 month before being used for the luciferase activity assay.

**EOD-FR treatment.** For transcript analysis, 8-day-old seedlings or 3-week-old adult plants grown under continuous WL conditions were treated with FR light (30 μmol m$^{-2}$ s$^{-1}$) for 15 min at the end of the light period (EOD-FR treatment). Seedlings and adult plants were treated for 7 days and harvested at the given time points (0, 0.5, 1, 3, 6 h) after the last treatment and frozen in liquid nitrogen for total RNA extraction.

For phenotypic investigation, 8-day-old seedling were transferred into soil and treated with EOD-FR as described above for 4 weeks until phenotypic investigation.

**Phenotypic investigation.** For phenotypic analysis, 2-week-old seedlings were grown under normal high R:FR (WL) or simulated shade (EOD-FR) for 4 weeks. After the main inflorescence became visible, the rosette leaves were detached, photographed, and counted. The blade area and petiole length of the first and second leaves were measured by ImageJ software (version 1.38) and averaged for blade area and petiole length, respectively. For rosette-leaf branch number, the rosette-leaf branches (longer than 0.3 cm) were counted and normalized to that of WT grown under WL conditions. For plant height, all the plants were measured at the time when growth of *35S::PIF5-OE* plants have ceased. Three biological replicates of each genotype were analyzed.

**Promoter analysis.** Promoter analysis was conducted using PLACE (http://www.dna.affrc.go.jp/PLACE/signalsacan.html). Promoters of *MIR156* genes (~3 kp upstream of mature *MIR156* sequence) were searched for the PBE-box (3′-CACATG-5′) or G-box (3′-CACGTG-5′) motifs.

**Plasmid construction and transformation.** For yeast one-hybrid assay, the coding regions of *PIF1*, *PIF3*, *PIF4*, and *PIF5* were PCR amplified and ligated to the pJG4-5 vector (Clontech, USA) at the *EcoR*I restriction site to produce AD-PIFs. Promoter fragment (~3 kb upstream of mature *MIR156* sequence) of each *MIR156* member and *SPL* member was PCR amplified and ligated to the pLacZi2μ vector[53] digested with *Sma*I to generate 2μ-pMIR156s and 2μ-pSPLs. For mutagenesis of the G-box or PBE-box in each 2μ-pMIR156 construct, primer pairs were designed according to Agilent Technologies (http://www.genomics.agilent.com) and used to generate plasmid-containing mutant G-box or PBE-box following the manufacturer's instructions.

For yeast two-hybrid assay, the coding regions of *PIFs* and *SPLs* were PCR amplified from cDNAs and ligated to the pGADT7 and pGBKT7 vectors (Clontech, USA) at the *EcoR*I restriction site to generate pGADT7-PIFs and pGBKT7-SPLs, respectively.

To generate the *pMIR156::GUS* constructs, all *MIR156* promoters were cloned into the vector pBI101 (Clontech, USA) at the *Sal*I site. All *pMIR156::GUS* constructs were then transformed into *Agrobacterium tumefaciens* strain GV3101 and further transformed into WT *Arabidopsis* (Col-0) using the floral dip method[54].

For luciferase assay, the coding region of *PIFs* was cloned into the SPYCE vector[55] at the *Sal*I restriction site to produce *35S::PIF-SPYCE*. All the *MIR156* promoters were amplified from individual WT 2μ-pMIR156s above and then ligated into the plasmid pGreenII0800-LUC (Biovector, USA) digested with *Sal*I to produce *pMIR156s::LUC*. The *MIR156* promoters harboring mutant G-box or PBE-box were generated using the corresponding same primer pairs with mutant 2μ-pMIR156s plasmids above as templates and inserted into the pGreenII0800-LUC vector at the *Sal*I site. The *35S::PIF-SPYCE* or *pMIR156::LUC* constructs were introduced into *Agrobacterium tumefaciens* strain EHA105 and these EHA105 cells harboring *35S::PIF5-SPYCE* or *pMIR156::LUC* were co-injected into *Nicotiana benthamiana* leaves.

All the primers used for the constructs above are shown in Supplementary Table 1.

**RNA extraction and quantitative RT-PCR.** Total RNA was isolated with Trizol reagent (Invitrogen, USA). For quantitative detection of the primary *MIR156s*, cDNA was firstly synthesized using M-MLV (Promega, USA) and detected with the TransStart Tip Green qPCR Super Mix (TransGen Biotech, China). For comparison of mature *MIR156*, total RNA was purified with the miRcute miRNA purification kit (Tiangen Biotech, China) and further cDNA was reverse transcribed with the miRcute miRNA first-strand cDNA synthesis kit (Tiangen Biotech, China). The mature *miR156* was detected with the miRcute miRNA qPCR Detection Kit (Tiangen Biotech, China). Quantitative RT-PCR was conducted with the SYBR Premix ExTaq kit (Takara, Japan) in a total volume of 25 μL on the Applied Biosystems 7500 real-time PCR system according to the manufacturer's manual. The level of *PP2A* (AT1G13320) transcript was adopted as an internal control. The expression of mature *MIR156* or each primary *MIR156* (*pri-MIR156A~H*) in different samples was calculated by the $2^{\Delta Ct}$ method, in which: $\Delta Ct = Ct_{(PP2A)} - Ct_{(target)}$. The expression level of target gene after treatments are the ratio of expression in treated samples compared with the controls by the $2^{\Delta\Delta Ct}$ method, in which $\Delta\Delta Ct = (Ct_{PP2A} - Ct_{target})_{treatment} - (Ct_{PP2A} - Ct_{target})_{control}$.

All the primers used for qRT-PCR above are shown in Supplementary Table 1.

**Yeast one-hybrid and two-hybrid assay.** To test the binding of PIF to *MIR156* or *SPL* promoter in yeast, the plasmids *AD-PIFs* and 2μ-pMIR156 (or 2μ-pSPL) were co-transformed into the yeast strain EGY48 mediated by PEG4000. The positive transformants grown on the SD/-Trp/-Ura medium (Clontech, USA) were transferred to the selection medium containing raffinose, galactose, and 5-Bromo-4-chloro-3-indolyl-beta-D-galactopyranoside (Amresco, USA) for blue color development. To quantify the β-galactosidase activity, the positive clones were cultured in liquid SD/-Trp/-Ura medium overnight, re-suspended in Z-buffer (40 mM NaH$_2$PO$_4$, 60 mM Na$_2$HPO$_4$, 1 mM MgSO$_4$, 10 mM KCl, pH 7.0, 50 mM β-mercaptoethanol), and then lysed with chloroform and 0.1% sodium dodecyl sulfate. Then 2-Nitrophenyl-β-D-galactopyranoside was added into the supernatants and incubated together at 28 °C for 5 min. The reaction was terminated by addition of 2 M Na$_2$CO$_3$ and the optical density of the supernatant was measured.

To examine possible interaction between PIF and SPL proteins, the plasmids pGADT7-PIFs and pGBKT7-SPLs were co-transformed into the yeast strain AH109 mediated by PEG4000. The positive transformants were first selected on the SD/-Ura/-Leu medium (Clontech, USA) and then transferred to SD/-Ura/-Leu/-His/-Ade selection medium (Clontech, USA) to detect the interaction. The interaction between pGADT7-T and pGBKT7-53 was used as a positive control.

**Recombinant protein production.** The truncated fragment of PIF5 encoding the bHLH domain (52 aa, Glu$_{259}$-Gln$_{310}$) was PCR amplified using the plasmid AD-PIF5 as the template and cloned into the vector GEX-4T-1 (Amersham, USA) at the *EcoR*I site to generate the pGEX-4T-PIF5 construct. The plasmid was transformed into the *Escherichia coli* Rosetta (DE3) strain. Expression of PIF5-GST fusion protein was induced by 0.4 mM isopropyl β-D-thiogalactopyranoside and incubation at 16 °C overnight. Cells were spinned down at 5000 rpm for 10 min and lysed by sonication. After centrifugation at 13,000 rpm for 40 min at 4 °C, the PIF5-GST fusion protein from supernatant was purified using glutathione-sepharose resin (GE Healthcare, USA). The eluted PIF5-GST fusion protein was dialyzed against the dialysis buffer (10 mM Tris-Cl, pH 8.0), aliquoted and stored at −80 °C until use.

**Gel mobility shift assay.** For DNA EMSA of PIF5 binding to promoters of *MIR156E* or *MIR156F*, two complementary 60-bp length oligonucleotides containing the G-box of *MIR156E* or *MIR156F* were synthesized and labeled with biotin separately. DNA probes were obtained by annealing two complementary oligonucleotides. DNA gel mobility shift assay was performed using the EMSA kit (Beyotime, China) following the manufacturer's protocol. Briefly, biotin-labeled probes were incubated for 20 min with the GST or GST-PIF5 protein in the binding buffer at room temperature. For competition reaction, 50× and 100× unlabeled cold probes were mixed with the labeled probes. The DNA–protein complex was separated by 5% native polyacrylamide gel electrophoresis and the signal of biotin was developed using the Biostep Celvin S420 system (Biostep, German).

**ChIP-PCR assay.** For ChIP-PCR assay, 8-day-old seedlings harboring *35S:PIF5-HA* were treated with EOD-FR for one time and kept in darkness for 3 h. Then the seedlings were cross-linked with 1% formaldehyde and the chromatin complexes were sonicated. The immuno complex was prepared following the method of Guo et al[56]. Briefly, after centrifugation at 14,000 × g for 30 min, the supernatant was pre-cleared with Protein-A-Agarose (Santa Cruz Bio, USA) and incubated at 4 °C for 1 h. After spinning, the supernatant was moved into a microtube and the HA-specific antibodies (Cali-Bio, USA) were added (for negative control, no antibodies were added). After incubation at 4 °C for 3 h, the Protein-A-Agarose was added and incubated for 2 h. After washing with 1 mL of each of the following buffers:

Buffer A (50 mM HEPES-KOH, pH 7.5, 1 mM EDTA, 140 mM NaCl, 0.1% Nadeoxycholate, 1% Triton X-100, 1 mM PMSF), Buffer B (50 mM HEPES-KOH, pH 7.5, 1 mM EDTA, 500 mM NaCl, 0.1% NaDeoxycholate,1% Triton X-100, 1 mM PMSF), Buffer C (10 mM Tris-HCl, pH 8.0, 0.5% NP-40, 0.5% NaDeoxycholate, 250 mM LiCl, 1 mM EDTA), and Buffer D (10 mM Tris-HCl, pH 8.0, 1 mM EDTA), the immuno complex was eluted from the agarose beads. The precipitated DNA was then recovered and quantified using quantitative PCR with their individual primer pairs. The values were standardized to the input DNA to obtain the enrichment fold. *PP2A* was used as an internal control.

**Luciferase activity assay**. *N. benthamiana* leaves were co-injected with *35S::PIF5-SPYCE* and *pMIR156::LUC* and incubated at 25 °C for 2–3 days. To image the luciferase luminescence, the leaves were detached and sprayed with 20 mg mL$^{-1}$ potassium luciferin (Gold Biotech, USA) and incubated in darkness for 5 min. The luciferase luminescence from the infiltrated area was imaged using Night SHADE LB 985 system (Berthold, Germany) with a 30 s exposure time, 4 × 4 binning, slow readout, and high gain. Quantification of luciferase activity was carried out with IndiGO software (version 2.03.0) using average luminescence counts per second. Luciferase activity was also measured using the Dual-Luciferase Reporter Assay System (Promega, USA) following the manufacturer's protocol and the relative firefly luciferase activity was counted as the ratio of firefly to *Renilla* luciferase activity (LUC/REN) for each sample.

**GUS assay**. Eight-day-old *pMIR156::GUS* seedlings were submerged, evacuated, and kept in the GUS staining solution (2 mM K$_4$Fe(CN)$_6$, 2 mM K$_3$Fe(CN)$_6$, 1 mg mL$^{-1}$ x-Gluc, 10 mM EDTA, 0.1% Triton X-100, 100 mg mL$^{-1}$ chloramphenicol, 50 mM Na$_2$HPO$_4$-NaH$_2$PO$_4$, pH 7.0) at 37 °C for 12 h. Then the stained seedlings were decolorized and fixed in 70% (v/v) ethanol and photographed using a Zeiss dissecting microscope. GUS activity assay was performed following the method of Jefferson et al[57].

**Western blotting**. Eight-day-old *PIF* overexpression seedlings (*35S::PIF1-MYC, 35S::PIF3-MYC, 35S::PIF4-HA,* and *35S::PIF5-HA*) grown under WL or treated with EOD-FR for one time were harvested at 0, 0.5, 1, 3, 6 h. Protein extracts were prepared as describe in Al-Sady et al[58] and total protein was quantified with the Quick Start Bradford Protein Assay Kit (Bio-Rad, USA). The purified anti-MYC antibodies (M047-7, MBL, Japan) or anti-HA antibodies (561-7, MBL, Japan) with 1:2000 and 1:10000 dilutions, respectively, were used to detect PIF1 and PIF3, or PIF4 and PIF5, respectively. As an internal control, the β-tubulin monoclonal antibodies (As10680, Agrisera, Sweden) at a dilution of 1:1000 were used with the goat anti-rabbit secondary antibodies (458, MBL, Japan) at a dilution of 1:8000. The original blot images were provided in Supplementary Fig. 14.

**Statistical analysis**. All real-time PCR reactions and other quantitative analysis were repeated at least three times. To evaluate the significant differences among the various genotypes treated or untreated with EOD-FR, the method of Brady et al[59] about two-way analysis of variance (ANOVA) with interaction was adopted by using the aov functions implemented in the "stats" package of the R programming language (version 3.3.1). Also the TukeyHSD in the "stats" package was used for all pairwise comparisons, with *p*-values corrected for multiple comparisons to control against type I errors. The ANOVA results were shown in Supplementary Data 1. The Student's *t*-test was adopted to analyze the significant differences between two groups.

**Data availability**. All relevant data that support the findings of this study are included within the article or available from the authors upon request.

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

## Acknowledgements

We thank Dr. Rongcheng Lin (Institute of Botany, Chinese Academy of Sciences) for kindly providing the transgenic seeds of *35S::PIF1-MYC* and *35S::PIF3-MYC*; Dr. Jia-qiang Sun (Institute of Crop Science, Chinese Academy of Agricultural Sciences) for providing the transgenic seeds of *35S::PIF4-HA* and *35S::PIF5-HA*. This work was supported by grants from National Natural Science Foundation of China (31570191) and Beijing Natural Science Foundation (5162024).

## Author contributions

H.W. and Y.X. designed the research; Y.X., Y.L., X.M., B.W., G.W., and H.W. performed the research and analyzed the data; H.W. and Y.X. wrote the paper.

## Additional information

**Competing interests:** The authors declare no competing financial interests.

