## [Peer Review File · Nature Communications]

Reviewers' comments:

Reviewer #1 (Remarks to the Author):

Review of Xie et al for Nature Communications.

In this manuscript Xie and colleagues reveal that the PIF proteins, transcription factors acting downstream of Phytochrome photoreceptors are regulators of MIR156 genes, an important micro RNA involved in multiple aspects of plant development. Both genetic and molecular techniques are used to make this connection. Connecting these two pathways will be of interest to many in the plant biology community.

There are several issues that need to be addressed before publication.

Major:

1) Statistical analyses. The phenotypic experiments are aimed at showing that various mutant lines have a different response to end-of-day far-red (EOD-FR) than wild type. Two types of analyses are performed: a) Student's t-tests separately for each genotype to determine if the genotype did or did not have a significant response to EOD-FR; and b) Student's t-tests on the percent reduction in response to shade, comparing each mutant back to wild type. The test listed as "a)" above is reasonable but does not address the question of significant differences among genotypes. However, I am concerned that the test listed as "b)" was not done correctly (at a minimum it was not documented well). Calculating the error of ratios between two different groups is quite challenging (both the numerator and denominator measurements have their own errors, propagating these through the division is non-trivial) and there is no indication of how this was dealt with (if at all). One correct approach would be to forgo the percentages and analyze the untransformed data with a 2-way ANOVA with interaction; the interaction term would indicate a significantly different response from wild-type. (For a partial discussion of this approach, see Brady et al, Plant Cell (2015) 27:2088-2094). If the ratio test have been done correctly, that detail needs to be presented in detail in the methods. Additionally Student's t-test is incorrectly referred to in the methods as "Student's t-test for analysis of variance test". Student's test is a test for similarity of means it is not an analysis of variance. It is unclear how Fisher's least significant differences were applied here; please explain.

2) The term "simulated shade" is incorrectly used. Since canopy shade lowers the R:FR ratio but does not cause an EOD-FR, it is incorrect to refer to the EOD-FR treatment here as "simulated shade". You can explain that EOD-FR causes plant phenotypes similar to that seen in shade or low R/FR and that you are therefore using it as a proxy, but then refer to your treatment as EOD-FR throughout.

Minor.

In the intro, when discussing the lack of studies of adult SAS, the work of Nozue et al (2015) should be acknowledged.

Fig 4: "CK" should be defined in figure legend.

Occasional editing for proper grammar is needed, but generally the writing is excellent.

Reviewer #2 (Remarks to the Author):

miR156 is an evolutionary conserved miRNA in both sequence and function in plants. It was first shown to be the master regulator of vegetative phase change in plants. Recent studies also demonstrated the versatile roles of miR156 in a plethora of different physiological, biochemical, and developmental processes in plants. Xie, et al. in this manuscript nicely demonstrated a brand-new role of miR156 by providing a molecular link between PIFs and miR156 in mediating shade avoidance syndrome in Arabidopsis. Results from this manuscript may have great potential in breeding for crops optimal for high density planting given that miR156 targets have been shown to be required for ideal plant architecture establishment in crops. The paper was well-written, data shown are solid and pretty much support what the authors claimed in the manuscript. I think this manuscript is well suitable for the publication in NC. The following is my questions for the authors:

1. Results in Fig1 and Fig.2 indicated that PIF-OE plants, pifq, MIR156-OE, MIM156, and wild type exhibited various responses, or even to a lesser degree, to simulated shade conditions. Therefore PIFs and miR156 actually are genetically not required for the response to EOD-FR. Is it possible that PIFs actually act in parallel to miR156, i.e. PIFs and miR156 share common downstream targets, say SPLs, to regulate the response given the proven result that SPL9, one target of miR156, physically interacts with DELLA, and DELLA also physically interacts with PIFs. Please include this possibility in the discussion.

2. As shown in Fig.3A, there was about 3-fold reduction in the level of mature miR156 in wild type after 3-hour treatment with EOD-FR. It is widely known that miR156A and miR156C constitute the majority of the mature miR156 in Arabidopsis. However, the authors were not able to show the interaction of PIFs with these two important loci. Please explain this.

3. According to the model presented in Fig. 5, PIFs act as negative regulators of miR156, and miR156 acts downstream of PIFs in response to simulated shade conditions; therefore we would expect that MIR156-OE plants would at least partially reduce the constitutive SAS conferred by PIF-OE plants; however data of PIF5-OE/ MIR156-OE plants in Fig.5A, 5B, and 5C did not support this hypothesis. Please explain this inconsistency in detail.

Below are our specific responses to each of the questions from the reviewers:

Reviewer #1 Remarks to the Author

Q1. Statistical analyses. The phenotypic experiments are aimed at showing that various mutant lines have a different response to end-of-day far-red (EOD-FR) than wild type. Two types of analyses are performed: a) Student's t-tests separately for each genotype to determine if the genotype did or did not have a significant response to EOD-FR; and b) Student's t-tests on the percent reduction in response to shade, comparing each mutant back to wild type. The test listed as "a)" above is reasonable but does not address the question of significant differences among genotypes. However, I am concerned that the test listed as "b)" was not done correctly (at a minimum it was not documented well). Calculating the error of ratios between two different groups is quite challenging (both the numerator and denominator measurements have their own errors, propagating these through the division is non-trivial) and there is no indication of how this was dealt with (if at all). One correct approach would be to forgo the percentages and analyze the untransformed data with a 2-way ANOVA with interaction; the interaction term would indicate a significantly different response from wild-type. (For a partial discussion of this approach, see Brady et al, Plant Cell (2015) 27:2088-2094). If the ratio test have been done correctly, that detail needs to be presented in detail in the methods.

Response: As suggested, two-way ANOVA with interaction as referred in Brady et al. (2015) was adopted in this study to better evaluate the significant difference among genotypes treated or untreated with EOD-FR. The aov functions implemented in the 'stats' package of the R programming language (version 3.3.1) were used. Also the TukeyHSD in the 'stats' package was used for all pairwise comparisons, with p-values corrected for multiple comparison to control against type I errors. We supplied the detailed outputs produced in the supplementary for your better consideration (see

Supplementary sheets) and re-marked the plots accordingly (See new Fig 1B, 2B, 5B, S1B, S3B and S11B).

In addition, we agreed with the reviewer that random variable expressed as the ratio of two random variables should be taken with care and accepted the reviewer's advice to remove the ratio (reduction or increase) and Fig 1C, 2C, 5C, S1C, S3C and S11C.

Q2. Additionally Student's t-test is incorrectly referred to in the methods as "Student's t-test for analysis of variance test". Student's test is a test for similarity of means it is not an analysis of variance. It is unclear how Fisher's least significant differences were applied here; please explain.

Response: Thanks for pointing this out. We corrected this mistake in the revised material and method section. We initially attempted to use the Fisher's least significant differences to measure the significant differences among genotypes. Later we found that the two-way ANOVA could better illustrate both the differences among genotypes and the differences for each genotype under WL and EOD-FR conditions (as shown in new Fig 1B, 2B and 5B). Thus we used the two-way ANOVA with interaction for this analysis.

Q3. The term "simulated shade" is incorrectly used. Since canopy shade lowers the R:FR ratio but does not cause an EOD-FR, it is incorrect to refer to the EOD-FR treatment here as "simulated shade". You can explain that EOD-FR causes plant phenotypes similar to that seen in shade or low R/FR and that you are therefore using it as a proxy, but then refer to your treatment as EOD-FR throughout.

Response: Thanks for the valuable suggestion! We revised the introduction of EOD-FR treatment in the result section as suggested.

Q4. In the intro, when discussing the lack of studies of adult SAS, the work of Nozue et al (2015) should be acknowledged.

Response: Thanks for kindly reminding! The work of Nozue et al. (2015) on adult SAS

was cited in the introduction section and the order of references was re-labeled accordingly.

Q5. *Fig 4: "CK" should be defined in figure legend.*

Response: Thanks. "CK" in this study means that the tobacco leaves injected with empty vector pSPYCE only, which was used as a control. For easier understanding all the "CK" in Fig 4 A and C were replaced with "pSPYCE".

Q6. *Occasional editing for proper grammar is needed, but generally the writing is excellent.*

Response: Thanks. We have carefully proofed the manuscript and corrected all the grammar mistakes we found.

Reviewer #2 Remarks to the Author

Q1. *Results in Fig1 and Fig. 2 indicated that PIF-OE plants, pifq, MIR156-OE, MIM156, and wild type exhibited various responses, or even to a lesser degree, to simulated shade conditions. Therefore PIFs and miR156 actually are genetically not required for the response to EOD-FR. Is it possible that PIFs actually act in parallel to miR156, i.e. PIFs and miR156 share common downstream targets, say SPLs, to regulate the response given the proven result that SPL9, one target of miR156, physically interacts with DELLA, and DELLA also physically interacts with PIFs. Please include this possibility in the discussion.*

Response: Thanks for the valuable comments. First, we wish to clarify that our results show that the *PIF-OE* plants exhibited constitute shade avoidance syndrome (SAS) even under normal high R:FR conditions (WL), and showed less sensitivity to EOD-FR treatment, while the *MIR156-OE* plants exhibited reduced SAS under normal WL conditions, and were more sensitive to EOD-FR treatment (Fig 1, Fig 2, Fig S1 and Fig S3). These observations suggest that the *MIR156-OE* plants had opposite SAS response

to the *PIF-OE* plants. Next we confirmed that PIFs directly bind to several *MIR156* promoters and repress their transcription (Fig 3 and Fig 4). Finally we verified the genetic relationship between *PIF* and *MIR156* by comparing the responses of various single mutants and higher order mutants to EOD-FR treatment (Fig 5; also see Response to Q3 below). Thus we concluded that *MIR156s* are direct targets of PIFs in mediating SAS.

Reviewer #2 also raised an intriguing possibility that PIFs may act in parallel to miR156, i. e. PIFs and miR156 share common downstream targets, say SPLs, to regulate the response given the proven result that SPL9, one target of miR156, physically interacts with DELLA, and DELLA also physically interacts with PIFs. Indeed, it has been previously shown that DELLA could directly interact with both SPL9 and PIFs, and suppress their activities (De Lucas et al., 2008; Feng et al., 2008; Yu et al., 2012). However, whether SPL proteins directly interact with PIFs remains unknown. To clarify this issue, we performed two sets of additional experiments (as suggested by you). First, we tested whether PIFs could directly bind to the promoters of SPL genes to directly regulate SPL expression (so that bypassing miR156s). Promoter sequence analysis found that the promoters of all of the miR156-targeted *SPLs* genes (including *SPL2*, *SPL3*, *SPL4*, *SPL5*, *SPL9*, and *SPL15*) except *SPL10* and *SPL11* contain the typical PIF binding sites (G-box or PBE-box) (Table 1 below for review only). However, yeast one-hybrid assay revealed that no direct binding of PIFs (PIF1, PIF3, PIF4 and PIF5) to any of the *SPL* promoters containing G-box or PBE-box (*pSPL2*, *pSPL3*, *pSPL4*, *pSPL5*, *pSPL9* and *pSPL15*) (Figure 1 below for review only). This result suggests that it is unlikely PIFs directly regulate the transcription of these *SPLs*. Second, we tested whether PIFs could regulate SPL activity by protein-protein interaction using yeast two-hybrid assay. Again, no interaction was detected between SPLs with PIFs (Figure 2 below for review only). These results are consistent with our proposition that PIFs regulate *SPL* gene expression/activity mainly through regulating *MIR156s*. We discussed this point in our revised manuscript.

Table 1. The number of G-box or PBE in putative *SPL* promoters

	Length	G-box (5'CACGTG3')	PBE (5'ACATG3')
pSPL2	3257 bp	1	2
pSPL3	2917 bp	1	2
pSPL4	3077 bp	1	2
pSPL5	2744 bp	1	0
pSPL9	3021 bp	0	5
pSPL10	1747 bp	0	0
pSPL11	2738 bp	0	0
pSPL15	1252 bp	0	1

Fig 1. No direct binding of PIFs to *SPL* promoters was detected.

Yeast one-hybrid assay showing that there was no direct binding of PIFs to the promoters of *miR156*-targeted *SPLs* genes. The binding of FHY3 to the *ELF4* promoter was used as a positive control (Li et al., 2011).

Fig 2. No interaction between PIFs and SPLs was detected.

Yeast two-hybrid assay showing that there was no interaction between PIFs and SPLs. Seven *miR156*-targeted SPLs were used in this assay. The interaction between pGADT7-T and pGBKT7-53 was used as a positive control.

Q2. As shown in Fig. 3A, there was about 3-fold reduction in the level of mature

miR156 in wild type after 3-hour treatment with EOD-FR. It is widely known that *miR156A* and *miR156C* constitute the majority of the mature *miR156* in *Arabidopsis*. However, the authors were not able to show the interaction of PIFs with these two important loci. Please explain this.

Response: Thanks for pointing out this. We have noticed that *miR156A* and *miR156C* played important roles in plant growth and development, especially in response to sugar promotion of vegetative phase change in *Arabidopsis* (Yang et al. 2013; Yu et al. 2013). Our search of the miRBase database showed that *miR156A-F* had similar content in *Arabidopsis*, i.e. *MIR156A*, *MIR156B*, *MIR156C*, *MIR156D*, *MIR156E* and *MIR156F* had comparable reads (10501, 10757, 10585, 12422, 10391 and 10418 reads per million in mature sequence, respectively), while *MIR156G* and *MIR156H* had rather low content (www.mirbase.org). Our promoter sequence analysis revealed that there were one PBE (PIF binding element, 5'CACATG3') in the *MIR156A* promoter and two G-boxes (5'CACGTG3', the typical binding site for PIF proteins) and one PBE in the *MIR156C* promoter (See supplementary Fig. 6). However, no binding of any PIF proteins (we used in this study) to these two promoters was detected in our yeast one-hybrid assay. Our ChIP-PCR results also showed there were no significant binding of PIFs to these fragments containing PBE or G-box elements. Besides, we searched all the data available online about PIF targets, including Leivar et al. (2009, 2012), Hornitschek et al. (2012) and Zhang et al. (2013). However, none of them detected *MIR156A* or *MIR156C* as putative direct targets of PIF proteins. Together, these results suggest that it is unlikely PIFs directly regulate *MIR156A* or *MIR156C*.

Q3. According to the model presented in Fig. 5, PIFs act as negative regulators of *miR156*, and *miR156* acts downstream of PIFs in response to simulated shade conditions; therefore we would expect that *MIR156-OE* plants would at least partially reduce the constitutive SAS conferred by *PIF-OE* plants; however data of *PIF5-OE/MIR156-OE* plants in Fig. 5A, 5B, and 5C did not support this hypothesis. Please explain this inconsistency in detail.

Response: Thanks for careful reviewing. We double checked all the figures and data

about the relationship between *PIF5* and *miR156* under both white light and EOD-FR conditions, including Figure 5A-B and Figure S11A-B. We believe that our data (rosette leaf number, leaf blade area, petiole length, rosette-leaf branches and plant height) are consistent with our proposition that *MIR156s* act downstream of *PIFs* in regulating *SAS*.

For example, in the case of rosette leaf number, under white light (WL) condition, wild type (WT), *PIF5-OE* and *MIR156-OE* plants had 11.66, 2.43 and 17.8 rosette leaves, respectively, while *PIF5-OE/MIR156-OE* plants had 12.45 rosette leaves, which showing that *MIR156-OE* largely rescued the effect conferred by *PIF5-OE* in the *PIF5-OE/MIR156-OE* plants (Fig 5A-B). Under EOD-FR treatment, the WT, *PIF5-OE* and *MIR156-OE* plants had 6.35, 1.98 and 14.2 rosette leaves, respectively, while the *PIF5-OE/MIR156-OE* plants had 8.47 rosette leaves, again showing that *MIR156-OE* largely rescued the effect conferred by *PIF5-OE* in the *PIF5-OE/MIR156-OE* plants (Fig 5 A-B). This consistency was also confirmed by the reduction (percentage) of rosette-leaf number. After EOD-FR treatment WT, *PIF5-OE* and *MIR156-OE* plants showed $43.5 \pm 4.1\%$, $17.8 \pm 2.7\%$, $30.2 \pm 2.9\%$ reductions, respectively, while *PIF5-OE/MIR156-OE* plants showed $31.9 \pm 4.8\%$ reduction, indicating that *MIR156-OE* largely compensated the effect of *PIF5-OE* in the *PIF5-OE/MIR156-OE* plants on rosette leaf number (Fig 5 A-B).

In the case of leaf blade area, under WL conditions, the leaf blade areas of WT, *PIF5-OE*, *MIR156-OE*, *PIF5-OE/MIR156-OE* were 0.186, 0.116, 0.365, and 0.126 cm², respectively; under EOD-FR conditions, the leaf blade areas of WT, *PIF5-OE*, *MIR156-OE* and *PIF5-OE/MIR156-OE* plants were 0.158, 0.109, 0.278 and 0.094 cm², respectively (Fig 5 A-B). Although the leaf blade area of *PIF5-OE/MIR156-OE* plants was more similar to that of *PIF5-OE* under WL conditions, however, after EOD-FR treatment, *PIF5-OE/MIR156-OE* displayed a similar reduction (percentage) in leaf blade area to *MIR156-OE*, but not *PIF5-OE*. The reductions for WT, *PIF5-OE* *MIR156-OE* and *PIF5-OE/MIR156-OE* plants were $15.1 \pm 3.6\%$, $5.8 \pm 2.8\%$, $23.7 \pm 2.5\%$, $25.4 \pm 2.6\%$, respectively (Fig 5 A-B).

In the case of petiole length, under WL conditions, the petiole lengths of WT, *PIF5-OE*, *MIR156-OE* and *PIF5-OE/MIR156-OE* plants were 0.672, 0.899, 0.534 and 0.753 cm, respectively; under EOD-FR conditions, the petiole lengths of WT, *PIF5-OE* and *MIR156-OE* and *PIF5-OE/MIR156-OE* plants were 0.773, 0.951, 0.679 and 0.943 cm, respectively. Although the petiole length of *PIF5-OE/MIR156-OE* plants was more similar to that of *PIF5-OE*, however, after EOD-FR treatment, *PIF5-OE/MIR156-OE* displayed a similar increase (percentage) to *MIR156-OE*, but not *PIF5-OE*. The increases for WT, *PIF5-OE*, *MIR156-OE* and *PIF5-OE/MIR156-OE* plants were $20.9 \pm 2.4\%$, $5.8 \pm 3.3\%$, $27.1 \pm 3.8\%$ and $26.2 \pm 3.7\%$, respectively (Fig 5 A-B), showing that *MIR156-OE* could largely rescue the effect of *PIF5-OE* in the *PIF5-OE/MIR156-OE* plants on petiole length in response to EOD-FR.

In the case of rosette-leaf branch, under WL conditions, the WT, *PIF5-OE*, *MIR156-OE* and *PIF5-OE/MIR156-OE* plants had 4.16, 0.37, 5.47 and 3.83 branches, respectively; under EOD-FR conditions, the WT, *PIF5-OE*, *MIR156-OE* and *PIF5-OE/MIR156-OE* plants had 1.89, 0.22, 2.06 and 1.65 branches, respectively (Fig 11 A-B). In response to EOD-FR, the WT, *PIF5-OE*, *MIR156-OE* and *PIF5-OE/MIR156-OE* plants exhibited $50.7 \pm 2.5\%$, $21.2 \pm 3.7\%$, $62.3 \pm 2.5\%$, $57 \pm 1.7\%$ reduction in rosette-leaf branches, respectively (Fig 11 A-B). These data suggest that *MIR156-OE* effectively rescued the effect conferred by *PIF5-OE* in *PIF5-OE/MIR156-OE* plants on rosette leaf branch number.

In the case of plant height, under WL conditions, the plant heights of WT, *PIF5-OE*, *MIR156-OE* and *PIF5-OE/MIR156-OE* plants were 16.32, 22.92, 7.06 and 6.45 cm, respectively; under EOD-FR conditions, the plant heights of WT, *PIF5-OE*, *MIR156-OE* and *PIF5-OE/MIR156-OE* plants were 19.4, 24.98, 8.21 and 8.47 cm, respectively (Fig 11 A-B). In response to EOD-FR, the plant heights of the WT, *PIF5-OE*, *MIR156-OE* and *PIF5-OE/MIR156-OE* plants increased by $20.1 \pm 3.9\%$, $8.9 \pm 2.3\%$, $31.2 \pm 2.9\%$ and $25.3 \pm 3.2\%$, respectively (Fig 11 A-B). These data indicated that *MIR156-OE* also effectively rescued the effect of *PIF5-OE* in the *PIF5-OE/MIR156-OE* plants on plant height.

References

- Brady, S. M. et al. (2015). Reassess the t test: Interact with all your data via ANOVA. *Plant Cell* **27**, 2088–2094 (2015).
- de Lucas, M. et al. A molecular framework for light and gibberellin control of cell elongation. *Nature* **451**, 480–484 (2008).
- Feng, S. et al. Coordinated regulation of *Arabidopsis thaliana* development by light and gibberellins. *Nature* **451**, 475–479 (2008).
- Hornitschek, P. et al. Phytochrome interacting factors 4 and 5 control seedling growth in changing light conditions by directly controlling auxin signaling. *Plant J.* **71**, 699–711 (2012).
- Leivar, P. et al. Definition of early transcriptional circuitry involved in light-induced reversal of PIF-imposed repression of photomorphogenesis in young *Arabidopsis* seedlings. *Plant Cell* **21**, 3535–3553 (2009).
- Leivar, P. et al. Dynamic antagonism between phytochromes and PIF family basic helix-loop-helix factors induces selective reciprocal responses to light and shade in a rapidly responsive transcriptional network in *Arabidopsis*. *Plant Cell* **24**, 1398–1419 (2012).
- Li, G. et al. Coordinated transcriptional regulation underlying the circadian clock in *Arabidopsis*. *Nat Cell Biol.* **13**, 616–622 (2011).
- Nozue, K. et al. Shade Avoidance Components and Pathways in Adult Plants Revealed by Phenotypic Profiling. *PLoS Genet.* **11**, e1004953 (2015).
- Yang, L. et al. Sugar promotes vegetative phase change in *Arabidopsis thaliana* by repressing the expression of MIR156A and MIR156C. *Elife* **2**, e00260(2013).
- Yu, S. et al. Sugar is an endogenous cue for juvenile-to-adult phase transition in plants. *Elife* **2**, e00269 (2013).
- Yu, S. et al. Gibberellin regulates the *Arabidopsis* floral transition through miR156-targeted SQUAMOSA PROMOTER BINDING–LIKE Transcription Factors. *Plant Cell* **24**, 3320–3332 (2012).
- Zhang, Y. et al. A quartet of PIF bHLH factors provides a transcriptionally centered signalling hub that regulates seedling morphogenesis through differential

expression-patterning of shared target genes in *Arabidopsis*. *PLoS Genet.* **9**, e1003244 (2013).

REVIEWERS' COMMENTS:

Reviewer #1 (Remarks to the Author):

My concerns have been adequately addressed. This is an interesting paper that should be of interest to a broad range of plant biologists.

Reviewer #2 (Remarks to the Author):

Xie, et al. in this manuscript nicely addressed my questions in a perfect manner. Results from this manuscript will have some valuable potential in breeding for crops optimal for high density planting. I highly recommend its acceptance by Nature Communications.